# The Contributions of Cancer-Testis and Developmental Genes to the Pathogenesis of Keratinocyte Carcinomas

**DOI:** 10.3390/cancers14153630

**Published:** 2022-07-26

**Authors:** Brandon Ramchatesingh, Jennifer Gantchev, Amelia Martínez Villarreal, Raman Preet Kaur Gill, Marine Lambert, Sriraam Sivachandran, Philippe Lefrançois, Ivan V. Litvinov

**Affiliations:** 1Division of Experimental Medicine, McGill University, Montreal, QC H4A 3J1, Canada; brandon.ramchatesingh@mail.mcgill.ca (B.R.); jennifer.theoret@mail.mcgill.ca (J.G.); amelia.martinezvillarreal@mail.mcgill.ca (A.M.V.); sriraam.sivachandran@mail.mcgill.ca (S.S.); 2Cancer Research Program, Research Institute of the McGill University Health Center, Montreal, QC H4A 3J1, Canada; raman.gill@mail.mcgill.ca (R.P.K.G.); marine.lambert@mail.mcgill.ca (M.L.); 3Division of Dermatology, McGill University Health Center, Montreal, QC H4A 3J1, Canada; philippe.lefrancois2@mcgill.ca

**Keywords:** basal cell carcinoma, cutaneous squamous cell carcinoma, cancer/testis antigen, embryonic stem cell, oncofetal antigen, precision oncology, biomarkers

## Abstract

**Simple Summary:**

In addition to mutations, ectopically-expressed genes are emerging as important contributors to cancer development. Efforts to characterize the expression patterns in cancers of gamete-restricted cancer-testis antigens and developmentally-restricted genes are underway, revealing these genes to be putative biomarkers and therapeutic targets for various malignancies. Basal cell carcinoma (BCC) and cutaneous squamous cell carcinoma (cSCC) are two highly-prevalent non-melanoma skin cancers that result in considerable burden on patients and our health system. To optimize disease prognostication and treatment, it is necessary to further classify the molecular complexity of these malignancies. This review describes the expression patterns and functions of cancer-testis antigens and developmentally-restricted genes in BCC and cSCC tumors. A large number of cancer-testis antigens and developmental genes exhibit substantial expression levels in BCC and cSCC. These genes have been shown to contribute to several aspects of cancer biology, including tumorigenesis, differentiation, invasion and responses to anti-cancer therapy.

**Abstract:**

Keratinocyte carcinomas are among the most prevalent malignancies worldwide. Basal cell carcinoma (BCC) and cutaneous squamous cell carcinoma (cSCC) are the two cancers recognized as keratinocyte carcinomas. The standard of care for treating these cancers includes surgery and ablative therapies. However, in recent years, targeted therapies (e.g., cetuximab for cSCC and vismodegib/sonidegib for BCC) have been used to treat advanced disease as well as immunotherapy (e.g., cemiplimab). These treatments are expensive and have significant toxicities with objective response rates approaching ~50–65%. Hence, there is a need to dissect the molecular pathogenesis of these cancers to identify novel biomarkers and therapeutic targets to improve disease management. Several cancer-testis antigens (CTA) and developmental genes (including embryonic stem cell factors and fetal genes) are ectopically expressed in BCC and cSCC. When ectopically expressed in malignant tissues, functions of these genes may be recaptured to promote tumorigenesis. CTAs and developmental genes are emerging as important players in the pathogenesis of BCC and cSCC, positioning themselves as attractive candidate biomarkers and therapeutic targets requiring rigorous testing. Herein, we review the current research and offer perspectives on the contributions of CTAs and developmental genes to the pathogenesis of keratinocyte carcinomas.

## 1. Introduction

Throughout the course of malignant progression, epigenetic alterations modify the transcriptome, allowing lineage- and developmentally-restricted genes to be transcribed ectopically [1,2,3]. Ectopically-expressed genes have diverse impacts on cancer cell biology. The most intuitive consequence of ectopic gene activation is a deregulation of cell differentiation pathways [3]. Activation of genes that are commonly restricted to prenatal development, including embryonic stem cell and fetal genes, is a hallmark of cancer cell de-differentiation to progenitor-like states and signifies differentiation impairment [4,5,6,7,8,9,10]. When lineage-specific genes are expressed ectopically, trans-differentiation to an alternative cell lineage may be observed [6,7,8,9,10,11]. Additionally, the functions of ectopically-expressed genes may be recaptured in malignant contexts [12,13]. The expression of lineage- and developmentally-restricted genes may influence additional aspects of disease biology, potentially promoting neoplasia, metastasis, response to therapies and other outcomes [14,15]. Investigating the phenotypic consequences of ectopic gene reactivation in cancer offers new opportunities for biomarker discovery and the development of precision-targeted therapies. 

Cancer cells evolve to express genes that are favorable to their growth, survival and dissemination. Reactivation of genes restricted to early mammalian ontogenesis confers selective advantages to cancer cells [14]. Reproductive germ cell and placental genes are the most common lineage-specific genes that are ectopically expressed in non-germ cell cancers [1]. Gametogenic and placental genes that can be ectopically expressed in non-germ cell cancers are called cancer-testis genes, or cancer-testis antigens (CTA) [14]. CTA expression represents a partial trans-differentiation of cancer cells towards gamete or placental phenotypes [11]. Gametogenic and placental cells possess phenotypic parallels to cancer cells including high proliferation rates, impaired differentiation, migratory phenotypes, genomic instability and immunological privilege [15]. These phenotypes are often attributable to the endogenous functions of CTAs. Over 200 CTAs have been documented in the literature [15]. CTAs can be classified as X or non-X chromosome linked and further subdivided into various subfamilies. Their pathological relevance has been studied in a plethora of malignancies, including skin, lung, breast, gastrointestinal, genitourinary and hematological cancers [7,8,9,10,16]. Beyond their relevance to cancer immunology, ectopic CTA expression influences cancer cell biology, as their natural functions are often recaptured in a novel way to promote carcinogenesis. 

Cancer cells can additionally benefit from adopting genes from the developmental transcriptome: embryonic stem cell (ESC) and oncofetal genes [5]. ESCs constitute the inner cell mass after fertilization and formation of a blastocyst [17]. These cells exhibit high proliferation rates, replicative immortality and pluripotency [18]. Some ESC genes orchestrate gastrulation and the subsequent stages of histogenesis and organogenesis. As a collective, ESC genes coordinate developmental dynamics in the early embryo. Perturbing their expression has significant consequences. ESC genes that are re-expressed in cancer include the core transcription factors (Sox2, Oct-4, and Nanog) that are necessary for pluripotency and cell survival as well as factors that regulate epithelial mesenchymal transition (EMT), the first step towards cancer cell migration and invasion [5,10,18,19]. Other developmental genes may also be expressed during fetal development (after the establishment of functional organ systems) and re-expressed in cancer. These genes are termed oncofetal antigens [20]. Fetal proteins coordinate diverse functions during development, including cell adhesion, EMT and growth [21]. Expression of ESC and oncofetal antigens in cancer cells promotes deleterious phenotypes. In particular, developmental genes are associated with the emergence of cancer stem cells (CSCs) [5,22]. CSCs are subpopulations of malignant cells with the ability to self-renew and spawn differentiated progeny [23]. These subpopulations are believed to contribute to disease recurrence, clonal evolution, dissemination and heterogeneity [23]. ESC genes can also modulate other aspects of neoplastic biology beyond the CSC hypothesis, including proliferation, metabolism, dissemination and therapeutic responses. Investigating how misappropriated CTA and developmental genes influence malignant progression may unveil new prognostic, diagnostic or therapeutic avenues for future studies. 

There is a persistent need to identify novel biomarkers and new therapeutic targets to facilitate diagnosis, infer clinical disease prognosis and treat keratinocyte carcinomas (KC). KCs are malignancies that arise from epidermal keratinocytes. Basal cell carcinoma (BCC) and cutaneous squamous cell carcinoma (cSCC) are the two types of KC, representing the first and fifth most prevalent cancers, respectively, worldwide [24,25,26,27,28]. BCC arises from keratinocytes in the basal layer of the interfollicular epidermis, mechanosensory niches and from hair follicle infundibulum stem cells [29]. Mutations in the Sonic Hedgehog (Shh)-Patched 1 (PTCH1) signaling axis are integral to BCC tumorigenesis [30]. Seventy to ninety percent of sporadic BCCs exhibit PTCH1 loss-of-function mutations and 10–30% exhibit smoothened gain-of-function mutations [31]. Risk factors for developing BCC include fair skin, intermittent ultraviolet radiation (UVR) exposure, prior ionized radiation, immunosuppression and certain genodermatoses, amongst other factors [32]. cSCC is characterized by malignant transformation of hair follicle bulge stem cells or interfollicular epidermal stem cells. Mutations in *TP53*, *CDKN2A*, *HRAS* and *NOTCH1* are among the most frequent drivers of cSCC tumorigenesis [33,34,35,36]. cSCCs may also be driven by human papilloma virus infection [32]. Risk factors for developing cSCC include fair skin, chronic UV exposure, immunosuppression (i.e., in solid organ transplant recipients receiving immunosuppressants) and select genodermatoses, among others [32,37]. cSCCs may develop in continuity with or progress from actinic keratoses (AK) [38]. In most cases, the prognoses of BCC and cSCC are relatively favorable. BCCs, in particular, are seldom aggressive or metastatic. Nonetheless, cases of locally invasive disease do occur and can be devastating/mutilating [39]. Risk of metastasis becomes significant for tumors that are ≥4 cm in diameter on the skin [40]. Discoveries of novel biomarkers to optimize diagnosis and risk stratification/prognostication remains critical in our ability to manage a large number of keratinocyte carcinomas. Ectopically expressed CTAs and developmental genes are emerging contributors to KC pathogenesis. Investigating the expression patterns and function of these genes in KCs may foster clinically relevant advances. This review outlines the expression and consequences of CTAs and early developmental genes for BCC and cSCC pathogenesis/progression. 

## 2. Cancer-Testis Antigens Are Widely Expressed in Keratinocyte Carcinomas 

The associations between CTA expression and the hallmarks of cancer are well-documented, and are continually illustrated by emerging research. The contributions of CTAs to KC pathogenesis are, unfortunately, largely unexplored. Walter et al. investigated the expression of 23 CTAs in human KC biopsies, using immunohistochemistry (IHC) and reverse-transcription quantitative polymerase chain reaction (RT-qPCR). Their panel included X-linked and non-X CTAs. At least one CTA was expressed in 81% of BCCs and 40% of cSCC tumors profiled in this study [41]. At least 40% of all KC tumors analyzed co-expressed two or more CTAs [41]. Relevant associations between CTA expression, disease histological subtypes and invasiveness were not established [41]. On average, BCCs co-expressed more CTAs per biopsy compared to cSCCs [41]. This finding was attributed to decreased cytotoxic T cell immunosurveillance in BCC tumors compared to cSCC tumors and to decreased tumor antigen presentation by BCC cells compared to cSCC cells [41]. BCC tumors contain fewer infiltrating cytotoxic T lymphocytes than cSCC tumors, and BCC cells express lower levels of MHC class I complexes compared to cSCC cells [41]. Hence, BCC cells are less prone to display tumor antigen epitopes that can be recognized by fewer cytotoxic T cells. 

In support of an alternative hypothesis, CTA expression was quantitatively and qualitatively compared between cSCCs from immunocompromised solid organ transplant recipients and immunocompetent patients [41]. Since immune status did not significantly impact CTA expression in cSCCs, it was concluded that immunosurveillance differences between BCC and cSCC cannot fully explain the differential expression of CTAs between BCC vs. SCC. Importantly, studies on the expression and function of specific CTAs in BCC and cSCC have been conducted. The CTAs known to be expressed in KC thus far may be grouped based on their functions. These groups include phenotypic regulation, invasion and stress response, and CTAs with unknown functions. 

### 2.1. CTAs That Regulate Cell Phenotype May Promote Neoplasia and Impair Differentiation in KC 

Precise spatiotemporal control of gene and protein expression are required throughout the process of gametogenesis. Unique transcriptional, post-transcriptional and post-translational regulatory mechanisms are engaged throughout gametogenesis and embryogenesis to orchestrate proper developmental dynamics and ensure timely transition in cell phenotypes [42]. These gene and protein expression regulatory factors can be recaptured in a novel way by cancer cells. The resulting modifications in cancer cell phenotypes promote carcinogenesis, notably proliferation and impaired differentiation. Herein we describe phenotype regulatory CTAs that are expressed in BCC and cSCC.

Preferentially expressed antigen of melanoma (PRAME) is a leucine-rich repeat protein ectopically expressed in a broad spectrum of cancers, including KC [41,43]. PRAME is the founding member of a multi-gene family of CTAs. In gametogenic cells, the PRAME protein maintains pluripotency and regulates proliferation [44,45,46]. In the presence of retinoic acid, PRAME binds retinoic acid receptors (RAR) at retinoic acid response elements (RARE) [44]. PRAME recruits EZH2 to the RARE, where EZH2 deposits trimethylation of histone 3 lysine 27 (h3k27me3), repressing transcription of RAR-target genes [44,47]. In addition to regulating transcription, PRAME regulates protein expression via Cullin-2, which targets cell-specific signaling factors for proteasomal degradation [45,48]. p14/ARF and Lin28 are among the cell signaling factors that PRAME targets for degradation [47,48,49]. In several types of cancer, PRAME expression correlates with aggressive disease characteristics, including high tumor grade, high proliferative index and metastatic propensity [43]. Walter et al. reported that PRAME was expressed in 55% of profiled KC tumors [41]. Immunostaining for PRAME protein was observed in malignant cells and in the dermis, with the strongest signals observed in acantholytic cSCC cells [41]. Corroborating these findings, a subsequent study reported positive PRAME expression in 86% of BCCs, 47% of poorly differentiated cSCCs and 20% of well-differentiated cSCCs in their study cohort [50]. All KC tumors, with the exception of one poorly-differentiated SCC, exhibited PRAME expression in 1–24% of tumor cells [50]. 

PRAME’s function, its correlation with clinicopathological parameters and therapeutic relevance may be important to KC treatment. PRAME’s activity as a retinoid signaling repressor makes it an attractive focus for future research on KC prevention and treatment. Via RAR signaling, retinoids normalize keratinization and promote epidermal turnover [51,52,53]. Systemic retinoids (e.g., acitretin) are effective chemoprophylactic therapies for AKs and KCs in immunosuppressed solid organ transplant recipients (SOTR) [52,54]. Retinoids are also used clinically to treat acute promyelocytic leukemia and pediatric neuroblastoma, and demonstrate potential as therapeutic agents for other cancers (e.g., squamous cell carcinomas) [55]. Considering the use of retinoids for KC chemoprophylaxis, there is a need to evaluate the contributions of ectopic PRAME expression/function to AK/KC pathogenesis and retinoid response in lesional skin. An adhesive patch pigmented lesion assay that probes PRAME and LINC00518 expression indicated a positive result in a small sample of actinic keratoses, and PRAME expression in premalignant tumors was previously described [56,57]. As a transcriptional repressor of retinoid signaling, PRAME expression may modulate sensitivity to retinoids in epidermal tumors. 

Whilst PRAME exerts transcriptional control of gene expression, insulin-like growth factor mRNA binding protein 3 (IMP3—also called IGF2BP3) exerts post-transcriptional gene regulation in gametes and ESCs [58]. The three members of the IMP family of mRNA binding proteins form ribonucleoprotein complexes that surround nascent mRNA transcripts [58]. IMPs protect against mRNA decay, prevent unnecessary translation, form transcript storage units and control intracellular trafficking of mRNA transcripts [58]. Ectopic expression of IMP3 in cancer correlates with adverse outcomes (e.g., higher tumor grade in bladder carcinoma, metastasis and poor survival in esophageal adenocarcinoma) [59]. Multiple studies have analyzed IMP3 expression in cutaneous neoplasms. One study conducted tissue microarrays of multiple tumor types, including BCC and cSCC. Of 40 cSCC samples, half stained negative, 10% had weak staining, 20% moderate and 20% strong expression of IMP3 [59]. Among BCCs, 79% of tumors stained negative, 8% exhibited weak staining, 8% exhibited moderate staining and 5% stained strongly [59]. IMP3 mRNA is not detected in normal skin [60]. This study also found that while keratoacanthomas (tumors of low malignant potential) did not express appreciable levels of IMP3, IMP3 expression was detected in 19 of 33 cSCC samples [61]. 

IMP3 supports proliferation and disrupts differentiation in cSCCs. Using RT-qPCR, Kanzaki et al. demonstrated that IMP3 was highly expressed in the human cSCC cell lines HSC-1 and HSC-5, and in HaCaT immortalized lesional keratinocytes but not expressed in normal skin [60]. IMP3 siRNA knockdown in HaCaT and SCC cells resulted in decreased proliferation or migration [60]. Additionally, IMP3-positive skin tumors exhibited higher Ki-67 labelling indices, suggesting increased proliferation of these tumor cells [60]. The observed phenotypes in IMP3-expressing cells may be attributable to a number of signaling networks that are perturbed by IMP3. For instance, IMP3 binds and positively regulates expression of cyclin D1, D3 and G1 mRNA in multiple cancer cell lines, thereby sustaining cell proliferation [62]. 

CTAs can leverage the ubiquitin proteasome system to regulate protein expression in gametes [42]. As mentioned earlier, PRAME serves as a substrate-recognition subunit for the Cullin-2 E3 ubiquitin ligase [45]. PRAME orchestrates the degradation of p14/ARF, which supports proliferation [48]. Additionally, PRAME can direct degradation of Lin28 in testicular germ cell tumors—regulating a key pluripotency pathway [49]. Members of the Melanoma Antigen-A (MAGE-A) family of CTAs are widely expressed in KC, guiding proteolysis to promote cancer progression. MAGEs coordinate spatiotemporal localization of E3 ubiquitin ligases, substrate targeting and enzymatic activity [63]. There are over 40 members of the MAGE family, approximately two thirds of which are CTAs [63]. Type 1 MAGEs (MAGE-A, -B, and -C) are CTAs with restricted expression patterns, while Type 2 MAGEs are expressed in various somatic tissues [63]. There are 12 members of the MAGE-A subfamily [63]. MAGE-As are expressed primarily in spermatogonial stem cells, where they function to protect the genome from genotoxic and starvation stresses [64]. These proteins possess MAGE homology domains that bind to RING E3 ubiquitin ligases [63]. Targets of MAGE-RING complexes include substrates implicated in cell signaling and oncogenesis, such as AMP-activated Protein Kinase (AMPK) and p53 [65]. Ectopic expression of MAGE-As promote malignant phenotypes including hyperproliferation, invasiveness and resistance to glycolysis inhibitors [63,64]. Previous reports indicate that MAGE-A3, -A4, -A9, -A10 and -A12 are expressed in KC tumors and contribute to disease progression [41,66,67]. 

Abikhair et al. investigated the expression patterns and prognostic relevance of MAGE-A3 expression in cSCC tumors [68]. They reported that 9 of 24 cSCC tumors probed for MAGE-A3 expression exhibited positive immunostaining [68]. Of these nine, the majority exhibited poor differentiation and some developed metastases [68]. Hence, MAGE-A3 expression in cSCC correlates with poor prognosis. In a follow up study, MAGE-A3 was found to be enriched in poorly-differentiated tumors with perineural invasion (PNI) relative to moderately differentiated tumors [69]. In vitro human cSCC cell lines and in vivo mouse models were used to demonstrate that MAGE-A3 may govern cell cycle progression and tumor growth in cSCC by regulating cyclin B, D and E expression, and also by promoting invasion [69]. 

MAGE-A4 is expressed in 25% of KC tumors [41]. Muelheisen et al. performed a more exhaustive assessment of MAGE-A4 expression in epidermal tumors. In this study, patient immune status was accounted for, allowing comparisons between tumors excised from immunocompromised SOTRs and immunocompetent patients [66]. In addition to cSCCs, MAGE-A4 expression was detected in AKs, Bowenoid AKs and SCC in-situ (Bowens disease) samples [66]. Expression was most common in Bowenoid AKs (71% of tumors) from immunocompetent patients and lowest in Bowenoid AKs from SOTR (25% of tumor profiles) [66]. Immunoreactivity was comparable between tumor types. Expression levels and immunostaining patterns varied depending on immune status, though the study lacked the power to statistically validate the former observation [66]. Immunostaining for MAGE-A4 was strongest in the basal layer of the epidermis in all four tumor types [66]. Chen et al. also reported that high MAGE-A4 expression was correlated with poorly differentiated cSCCs with PNI [69]. 

MAGE-A9 and -A10 are expressed in BCC and cSCC as well, but their functions and correlations with disease characteristics are unknown. MAGE-A9 was expressed in 23.9% of KC tumors, with a predilection for BCC [41]. MAGE-A10 was expressed in over 50% of tumor cells in roughly 32% of BCC cases [67]. This gene was expressed in less than 2% of cSCCs profiled [67]. Zhao et al. demonstrated that MAGE-A12 overexpression in cSCC tumors correlates with advancing age and high tumor grade [70]. Knockdown of MAGE-A12 in A431 cSCC cells decreased invasion and reduced tumor growth in a xenograft mouse model [70]. Overall, MAGE proteins are associated with proliferation and impaired differentiation of KC tumors, and may bear prognostic and therapeutic relevance. Although MAGE-A proteins are expressed in BCC, their functions are not well characterized. Additional members of the type 1 MAGE family of CTAs remain to be profiled in KC.

The aforementioned gene and protein regulatory CTAs are expressed in KC tumors. Gametes use these factors to support multiple phenotypes, including high proliferation rates and impaired differentiation (Figure 1). Likewise, in cancer, these CTAs can regulate proliferation and differentiation. PRAME, MAGE-As and IMP3 regulate expression of cell cycle regulatory factors (e.g., p14, p53, cyclins, etc.) In KC cells, IMP3 and MAGE-A regulate proliferation. Regarding differentiation, PRAME is more often expressed in poorly differentiated compared to well-differentiated cSCCs. Likewise, MAGE-A expression correlates with poorly differentiated cSCCs. Facilitating migratory phenotypes in cSCC cells may implicate IMP3 in EMT—a manifestation of cellular plasticity and a documented consequence of IMP3 expression [71,72]. Based on CTA expression patterns and functions in KC, we provide evidence for the noteworthy contributions of CTAs to neoplastic biology. Reflecting the needs of gametogenic cells, CTA expression can promote aggressive phenotypes in cancer. Additional studies are needed to further characterize these outcomes and their clinical implications. 

### 2.2. CTAs May Influence Invasion and Stress Response Phenotypes in KC

Cancer cells and gametogenic cells are attuned to navigate and thrive in their microenvironments. These cells express genes that regulate their motility and their capacity to withstand and adapt to threats to their viability, such as reactive oxygen species (ROS), genotoxic and metabolic stresses. Genes that control motility and stress response in gametogenic cells can be misappropriated by cancer cells to support their survival, migration and increase their fitness. Here, we describe CTAs that are ectopically expressed in KC and influence invasiveness and stress response phenotypes in this malignancy. 

CTAs can alter invasive phenotypes in cancer cells by modulating their interactions with the extracellular matrix (ECM). Testis Expressed 101 (TEX101) is a membrane-anchored glycoprotein that is required for sperm capacitation (the process whereby spermatozoa become motile and competent to fertilize an oocyte) [73]. Studies probing TEX101 expression and function in cancer are lacking. Yin et al. demonstrated that TEX101 suppresses invasion and metastasis by de-activating multiple ECM degrading enzymes, including urokinase plasminogen activator and matrix metalloproteases (MMP) 2 and 9, which mediate ECM degradation, promoting invasion and metastasis [74]. Consequently, ectopic TEX101 expression may be a negative regulator of invasion. This hypothesis is supported by additional studies demonstrating negative correlations between TEX101 expression and metastasis [75]. Ghafouri-Fard et al. used IHC to investigate expression of three CTAs, including TEX101 in BCC tumors. In this study, 38% of BCC samples demonstrated positive staining for TEX101 protein, and such expression was associated with the nodular BCC subtype [76]. Provided the role of TEX101 in inhibiting invasion, characterizing the prognostic significance of TEX101 expression in BCC tumors is a fascinating prospect, considering that local invasion can be debilitating functionally and warrant more aggressive surgical/radiotherapy treatments in cosmetically sensitive areas. The possible expression and functions of TEX101 in cSCC have not been studied. 

Sperm Associated Antigen 9 (SPAG9--also known as JIP4 or JLP) is a member of the c-jun N-terminal Kinase interacting protein (JIP) family that is implicated in cancer cell invasion [77]. Widely expressed in KC tumors, this cell-surface scaffolding protein is endogenously expressed on the acrosomal compartment of spermatozoa, serving as a signaling mediator of spermatozoa-oocyte interactions via the p38 MAP kinase and the c-jun N-terminal Kinase (JNK) pathways [77]. Several type of cancers, including bladder, breast, prostate, renal and gastrointestinal, exhibit ectopic SPAG9 expression [77]. In these malignancies, ectopic SPAG9 expression is associated with high proliferation, resistance to ROS and heightened metastatic propensity [78,79,80]. SPAG9 has been shown to upregulate expression of MMP2 and MMP-9 proteins [81,82]. Mechanistically, it has been proposed that SPAG9 acts as a negative regulator of tissue-inhibitor of metalloproteinases (TIMP)-1/2, which are endogenous inhibitors of MMPs [81]. By liberating MMPs from TIMP regulation, SPAG9 activity can promote local invasion and metastatic spread [81]. Using IHC, Seleit et al. probed SPAG9 expression in KC tumors. This study reported that SPAG9 protein is expressed in 90% of BCCs and over 80% of cSCCs from their sample cohort [83]. Positive staining was observed in both tumor cells and in the dermis for BCCs but was restricted to cancer cells in cSCC [83]. Roles for SPAG9 in the pathogenesis of BCC and cSCC have not been investigated thus far. Given SPAG9’s documented impact on MMP expression, correlating expression with KC disease stage and prognosis may be informative. The influence of SPAG9 on invasion and therapy-induced stress response may also have implications for KC prognosis and treatment. Miao et al. reported that SPAG9 confers resistance to 5-fluorouracil (5-FU) treatment in gastric cancer [84]. Eke et al. demonstrated that SPAG9 attenuates cetuximab-induced radio-sensitization in head and neck SCC [85]. 5-FU can be used to treat AKs and select cSCCs [86,87]. Cetuximab and radiation are also used in advanced KCs [39,88]. Since SPAG9 has been shown in some cancers to regulate sensitivity to 5-FU, cetuximab and radiotherapy, evaluating the relevance of SPAG9 expression to the treatment of epidermal tumors could be informative. 

Akin to SPAG9, Testis Specific Gene A10 (TSGA10) is another CTA expressed in KC that may influence stress responses. TGSA10 is a sperm tail sheath protein that is necessary for sperm motility and proper mitochondrial organization in the midpiece of mature spermatozoa [89]. This protein binds and prevents nuclear localization of the transcription factor hypoxia inducible factor 1α (HIF-1α) in spermatids [90]. Ectopic TSGA10 is detected in a multitude of cancers and has been proposed as a diagnostic biomarker for select malignancies [91,92]. In cancer, TSGA10 has been shown to inhibit angiogenesis by sequestering HIF1-α [93,94,95]. TSGA10 expression is also associated with decreased cell migration, repression of EMT and decreased metastatic potential [95,96]. Evidently, TSGA10 serves tumor suppressor functions. Mobasheri et al. profiled TSGA10 expression in several tumor types using RT-qPCR [91]. This study concurrently profiled the expression of another CTA: synaptonemal complex 3 (SYCP3) [91]. While the SYCP3 gene was not expressed in the studied skin tumors, TSGA10 gene expression was detected in 40% of cSCCs and 75% of BCCs [91]. Functions of this gene in KC have not been explored.

In brief, CTAs that control local migration and stress response in gametes and cancer cells are ectopically expressed in BCC and cSCC. Few studies have investigated the functions of these genes in KC pathogenesis and meaningful correlations with clinicopathological features. Although extrapolating putative relevance based on previous research can generate valuable hypotheses, studies to directly assess the biological functions of these genes and whether their ectopic expression bears any clinical relevance in KC management are required. 

### 2.3. CTAs of Unknown Significance Are Expressed in KC and in Normal Epidermis

Other CTAs have been detected in KC cells, but their contributions to cancer biology are poorly understood (summarized in Table 1). Either their functions in gametes, their functions in cancer, or both have not been elucidated. As some of these genes exhibit considerable expression in KC tumors, evaluating their function is a compelling prospect.

Although this review encompasses ectopically expressed genes in KC tumors, it is worth mentioning the activities of CTAs expressed in non-transformed keratinocytes. Contrary to dogma, CTA expression is biased to gametes and cancer cells, but not restricted to them. Several CTAs, including brother of the regulator of imprinted sites (BORIS), piwi-like RNA mediated gene silencing 2 (PIWIL2) and heat shock protein 105 (HSP105) are expressed at low levels in healthy epidermal keratinocytes [97,98,99,100]. Their expression levels are altered in cancer (these CTAs are excluded from Table 1). BORIS and PIWIL2, which serve genome protective functions in germ cells, are expressed in normal skin but absent from BCC and cSCC tumors [41,100]. Loss of BORIS in keratinocytes is also linked to increased genomic instability and proliferation [100]. In contrast, HSP105, a stress-responsive chaperone, is overexpressed in poorly differentiated cSCC tumors and metastases [98,99]. These observations are interesting for at least two reasons. As the primary interface with the external environment, the skin is exposed to a multitude of insults, including but not limited to ultraviolet radiation (UVR), chemical stresses and pathogenic microorganisms. Baseline expression of CTAs may protect against transformation, consistent with the observation that genome protective factors such as PIWIL2 and BORIS are absent from KC tumors. On the other hand, overexpression of protective factors may foster more treatment-resistant cancer cells. HSP105 overexpression is observed in 100% of cSCC metastases and correlates with high tumor grade [98,99]. In vitro studies illustrate that HSP105 expression can confer a selective growth advantage to cSCC cells [98]. Studying the significance of CTA expression in non-neoplastic and pre-neoplastic epidermis may yield insights into designing targeted chemoprevention or in predicting cancer development. CTAs are an active area of interest for the development of precision therapeutics. However, the majority of these genes have yet to be characterized in KC tumors. 

## 3. Embryonic Stem Cell Genes Are Ectopically Expressed in Keratinocyte Carcinomas 

KC cells can recapture genes from the developmental transcriptome in novel ways to support carcinogenesis. ESC and oncofetal genes are associated with diverse phenotypic outcomes in BCC and cSCC. One outcome of interest is the emergence of CSC subpopulations. CSCs in KC tumors have been documented, and are associated with tumor initiation and stemness [101,102]. These subpopulations re-express early developmental genes including ESC and oncofetal genes. These genes have diverse roles in carcinogenesis and cancer progression. For the purposes of this review, these ectopically-expressed genes will be grouped based on their activities in prenatal development: elements of the core ESC transcriptional circuitry, EMT factors and oncofetal proteins

### 3.1. Core ESC Transcription Factors Support Multiple Hallmarks of Cancer in KC 

The core ESC transcription factors that are ectopically expressed in KC regulate tumor initiation, stemness and several other outcomes. The three core transcription factors are sex determining region Y-box protein 2 (Sox2), octamer binding transcription factor 4 (Oct-4) and Nanog [17]. Their functions during embryogenesis are reviewed extensively by Young et al. [17]. Effectively, these genes promote a stable pluripotent state by activating transcription of pluripotency genes and repressing lineage specific genes in ESCs [17]. Their expression is restricted to the ESCs in the inner cell mass; they are undetectable in adult tissues [17]. Ectopic activation of these factors in cancer correlates with aggressive disease and the emergence of CSC subpopulations [5]. Milosevic et al. profiled expression of the core ESC factors in primary BCC tumors using RT-qPCR [103]. Tumor tissues were enriched in Sox2, Oct-4 and Nanog mRNA transcripts [103]. Expression was lower in close surgical resection margin tissue (within 3 mm) and undetectable in adjacent normal tissues [103]. Cells from the tumor mass exhibited resistance to chemotherapies (5-FU and cisplatin) and increased stemness in vitro, suggestive of CSC populations [103]. This study demonstrated that ESC genes are enriched in BCCs and may support resistance phenotypes. Analogous studies for cSCC have not been conducted. Nonetheless, studies investigating the individual contributions of specific core ESC genes in BCC and cSCC pathogenesis have been performed, revealing that these transcription factors significantly impact carcinogenesis.

In ESCs, Sox2 forms a complex with Oct-4 that is necessary for establishing pluripotency [17]. Li et al. performed Sox2 knockdown and overexpression in primary human BCC cells. Transient Sox2 knockdown attenuated proliferation and migration [104]. Knockdown also increased expression of E-cadherin protein whilst decreasing fibronectin and vimentin [104]. Opposite effects were observed when Sox2 was overexpressed [104]. These findings implicate Sox2 expression in BCC cells with heightened proliferative and invasive phenotypes. These effects were found to be mediated by Serine Arginine Protein Kinase 1 (SRPK1), a kinase that is upregulated by Sox2 and promotes signaling via the PI3 kinase/AKT pathway [104]. This study illustrates a role for Sox2 in BCC pathogenesis and highlights the Sox2-SRPK1-PI3K/AKT cascade as a potential target for BCC treatment [104]. 

In cSCC, ectopically-expressed Sox2 is implicated in tumor initiation and malignant progression. Boumahdi et al. reported that Sox2 protein is expressed in 29 of 40 AKs and 25 of 39 invasive cSCC tumors [105]. Sox2 expression was also observed in murine papillomas (analogous to human AKs) and cSCCs tumors that were induced using the chemical carcinogen 7,12-Dimethylbenz[a]anthracene/12-O-tetradecanoylphorbol 13-acetate (DMBA/TPA) [105]. Isolated Sox2-expressing murine tumor cells had a greater tumor-initiating potential compared to Sox2-negative cells [105]. Roles for Sox2 in early-stage malignant progression were investigated. Deletion of Sox2 in established papillomas and cSCCs resulted in tumor regression [105]. Transcriptional profiling revealed that Sox2 expressing cells are enriched in genes involved in proliferation, adhesion and stemness [105]. Siegle et al. further elaborated the role of Sox2 in skin carcinogenesis. Sox2 emerged as a gene that is ectopically expressed in murine cSCC tumor initiating cells (TICs) but not in other progenitor cells of the skin [106]. Expression of Sox2 was detected at the tumor-dermal interface of primary murine and human cSCC specimens [106]. Sox2 also stimulates proliferation and production of pro-angiogenic factors in TICs, including expression of neuropilin-1/2 and secreted phosphoprotein 1 [106]. Conditional deletion of Sox2 in murine skin reduced tumor formation following treatment with DMBA/TPA [106]. These two studies outline critical roles for Sox2 in tumor initiation and stemness in cSCCs. A subsequent report found that paired like homeobox 1 (Pitx1), an ectopically expressed oncofetal antigen regulated by Sox2, represses squamous differentiation in SCC cells [107]. Mechanistically, Pitx1 cooperates with Sox2 and p63 in a positive feedback loop to repress Krüppel-like Factor 4 (Klf4). Klf4 controls epithelial stratification and late-stage squamous differentiation [108]. By repressing Klf4, the coordinated activity of Sox2 and p63 represses squamous/keratinocyte differentiation [107]. 

Another role for Sox2, as a factor involved in metabolic reprogramming, has been proposed. Hsieh et al. demonstrated that Sox2 coordinates with p63 to increase expression of GLUT1, a glucose transporter enriched in SCCs from various anatomical sites [109]. Enrichment of GLUT1 renders cSCC cells dependent on glucose as an energy source, potentially offering an opportunity for a low-glucose precision nutrition strategy to help optimize cSCC management, and revealing a strong correlation between glycaemia and poor prognosis in cSCC [109]. Based on the expression of Sox2 in KC tumors and the noteworthy contributions to the biology of these cancers, characterizing corollaries between Sox2 and clinical parameters is warranted. 

Oct-4, which associates with Sox2, also contributes to KC tumorigenesis. Conflicting results have been published regarding Oct-4 expression in KC tumors [103,110,111]. Katona et al. reported that Oct-4 protein is expressed in basal keratinocytes in healthy epidermis, but not retained in cutaneous tumors, including BCCs and cSCCs [110]. Conversely, Milosevic et al. detected Oct-4 mRNA expression in BCC tumors [103]. Adhikary et al. also isolated subpopulations of epidermal CSCs expressing Oct-4 from the cSCC cell line, SCC-13. Oct-4 is implicated in early tumor formation. Hochedlinger et al. investigated the effects of ectopic Oct-4 expression on mature murine tissues in vivo [112]. Oct-4 expression in murine epidermis resulted in the formation of dysplastic lesions with increased numbers of immature, Ki-67 positive cells and invasion into the subcutaneous tissue [112]. These lesions appeared to expand from the hair follicle outer-root sheath sub-compartment [112]. Supporting a role for Oct-4 in tumor formation, ectopic expression of Oct-4 in keratinocytes has also been implicated in stemness. Grinnel et al. performed an in vitro experiment, transfecting Oct-4 into mouse interfollicular keratinocytes [113]. Oct-4 expressing cells de-differentiated to progenitor stem cell states, marked by the expression of other ESC factors such as Sox2 and Nanog [113]. Understanding that Oct-4 expression can elicit these phenotypes in keratinocytes makes the prospect of characterizing its expression and regulation in healthy epidermis and premalignant lesions appealing. Oct-4 expression may be induced in keratinocytes under select circumstances. Transient Oct-4 activation in keratinocytes in vitro was shown to occur in response to treatment with the cancer drugs doxorubicin and decitabine [114]. Whether this induction occurs in patients treated with these drugs remains to be demonstrated. 

Although Oct-4 expression is detected in BCC, the functions of Oct-4 in BCC pathogenesis are elusive. Oct-4 expression and function have been investigated in cSCC. Adhikary et al. noted that Oct-4-expressing CSC subpopulations exhibited greater capacity for tumor formation in a xenotransplantation model [111]. Consistent with the CSC hypothesis, evidently, Oct4 may support stemness in cSCC cells. Mechanistic dissection of Oct-4 functions in KC tumors as well as more robust gene expression studies (including analysis of premalignant tissues) are needed.

Nanog co-occupies Sox2/Oct-4 binding sites and is required for sustaining pluripotency in ESC cells [17]. Nappi et al. probed Nanog’s contributions to KC pathogenesis using BCC and cSCC cell lines. Nanog expression in BCC tumors is validated by the results of this study. Overexpression of Nanog in BCC cells increased transcription of Type II deiodinase, which converts thyroxine (T4) pro-hormone to active triiodothyronine (T3) [115]. Thyroid hormones (TH) are involved in epidermal development and homeostasis [116]. They also contribute to the pathogenesis of KC, serving pro-oncogenic functions in BCC cells [117,118]. This study demonstrated that Nanog promotes BCC development by over-stimulating the thyroid hormone pathway, which in turn promotes early-stage tumor formation, EMT and migration in a genetically-engineered mouse model of BCC [115]. The functions of Nanog in cSCC are diverse. Nappi et al. reported that ectopic Nanog expression correlates with cSCC disease stage [115]. Additionally, Nanog was shown to coordinate with TH activity to promote cell migration and EMT [115]. Unexpectedly, Kim et al. uncovered that Nanog generates DNA damage in keratinocytes, which promotes cSCC development in a p53-null background [119]. In a transgenic mouse model with Nanog expression in the skin, thickened epidermis and aberrant hair follicle cycling was evident, and attributed to premature differentiation and appearance of Nanog-expressing epidermal stem cells [119]. The induction of differentiation was determined to be a consequence of p53 activation [119]. Investigating the stimulus for p53 activation, it was found that Nanog-overexpressing cells exhibit higher levels of DNA double stranded breaks, since Nanog impairs KAP1, a chromatin remodeling enzyme that preserves genomic stability [119]. When p53 was deleted and Nanog expressed in murine skin, spontaneous tumor formation was observed [119]. This study illustrated the potential for Nanog to induce tumorigenesis and genomically unstable cancers in a p53 null background. Palla et al. described a role for Nanog in cSCC tumor formation. Contrary to other studies, this report indicates that Nanog is expressed at low levels in basal-layer keratinocytes [120]. In this study, Nanog overexpression in cSCC was associated with proliferation of basal cells, impaired differentiation, secondary tumor formation and EMT [120]. As detailed above, Nanog expression contributes to multiple aspects of KC development. 

Additional embryo-specific factors that interact with the core pluripotency circuitry include Zinc-finger x-linked (Zfx), a transcription factor that maintains pluripotency in hematopoietic and embryonic stem cells and acts as a transcriptional co-factor of Oct-4 [121,122]. Palmer et al. demonstrated that Zfx is overexpressed in human BCC tumors [122]. Expression is also detected in cSCCs, AKs and in-situ cSCC [122]. Zfx expression was necessary for BCC tumorigenesis driven by Hedgehog pathway activation [122]. PTCH1-null mice were generated to model BCC and coupled to an inducible Zfx deletion [122]. When Zfx was concurrently knocked-down with PTCH1 in these mice, tumorigenesis was reduced [122]. Despite being detected in AK and cSCC tumors, the involvement of Zfx in the pathogenesis of these tumors has not been investigated. Given its apparent role in tumor initiation, investigating the targetability of Zfx in pre-malignant and malignant tumors, as well as possible correlations between expression and clinical parameters, are compelling prospects. 

In summary, core ESC transcription factors and their interactors can be repurposed by KC cells to support multiple hallmarks of cancer. Critical roles for ESC genes in tumorigenesis are described, as well as roles in cell plasticity, genomic instability and metabolic reprogramming. 

### 3.2. Ectopic Expression of EMT Factors in KC

EMT is a physiological process during embryogenesis that is necessary for the establishment of the three embryonic germ layers [123]. The induction of EMT in transformed tissues is an early event in invasion and metastasis and is the most recognized manifestation of phenotypic plasticity [3]. EMT is coordinated by a complement of embryo-specific transcription factors including Snail1, Slug, Twist1 and Zeb1/2 in both the embryo and transformed tissues [124]. While Snail1 and Slug are shown to be expressed at low to moderate levels in normal epidermis, the other factors are absent from normal adult skin and are re-expressed in malignant tissues, promoting invasive phenotypes [125]. 

The transcription factor Twist-related protein 1 (Twist1) is essential for mesoderm formation during embryonic development [126]. A role for Twist1 in KC is controversial due to paradoxical findings regarding its expression in healthy human skin [127]. Vand-Rajabpour et al. performed RT-qPCR on primary BCC biopsies and reported that Twist-1 mRNA is decreased in BCCs compared to normal skin [127]. It was hypothesized that this reduced Twist1 mRNA expression in BCC explains the low propensity for these cancers to become invasive [127]. Contrary to these findings, Beck et al. demonstrated that Twist1 protein is absent from normal epidermis, but is ectopically expressed in murine and human hyperplastic skin, pre-cancerous lesions and cSCCs [128]. A low level of Twist-1 protein was detectable in the dermis. Conditional deletion of Twist1 expression in murine epidermis attenuated DMBA/TPA-induced SCC formation [128]. cSCC cells with Twist1 knockdown demonstrate decreased tumor propagating potential [128]. Deletion of Twist1 in papillomas caused tumor regression without impacting markers of EMT, suggesting that Twist1 is necessary for tumor maintenance independent of EMT [128]. It was determined that Twist1 regulates p53 stabilization, which in turn enables and sustains skin tumors [128]. Hence, beyond EMT and invasion, Twist1 plays a crucial role in tumorigenesis, stemness and tumor maintenance. These results were corroborated by a subsequent study [129]. Eguiarte-Solomon et al. further found that conditional Twist1 deletion in murine epidermis attenuated UV-induced hyperproliferation and tumorigenesis [130]. Twist1 overexpression in keratinocytes also impaired keratinocyte differentiation in vitro [130]. These combined results indicate that Twist1 may be implicated in tumorigenesis and shifting the balance between proliferation and differentiation. 

Zinc Finger E-box-binding homeodomain 1 (Zeb1) is another EMT transcription factor that is associated with cSCC pathogenesis. Zeb1 is not expressed in normal epidermis, but is detectable in cSCCs [131]. Miro et al. reported that TH fuels the pathogenesis of cSCC by activating Zeb1 expression. To be specific, the type II deiodinase-TH signaling axis in DMBA/TPA-induced mouse cSCCs promotes invasive phenotypes and EMT transcriptional programmes via Zeb1 upregulation [131]. Zeb1 depletion in cSCC cells, using shRNA, attenuated migration and EMT induction caused by TH treatment [131]. Whether Zeb1 is expressed in BCC tumors or contributes to BCC pathogenesis has not been investigated.

### 3.3. Emerging Contributions of Oncofetal Genes to Early Skin Tumorigenesis

In addition to ESC genes, oncofetal genes were also shown to participate in KC tumorigenesis. High mobility group AT-hook 2 (HMGA2) is an oncofetal protein that binds DNA at select promoters/enhancers, remodeling chromatin architecture and serving as a scaffold for additional factors [132]. Ha et al. investigated HMGA2 expression in UV-induced skin carcinogenesis. RT-qPCR and immunofluorescence were used to demonstrate that HMGA2 is expressed in neonatal foreskin keratinocytes but not in adult epidermis [133]. HMGA2 is re-expressed in cSCC tumors, and in immortalized lesional HaCaT keratinocytes [133]. UVR upregulates HMGA2 expression in keratinocytes and in SCC cells in vivo [133]. Since HMGA2 is expressed in non-lesional and UV-treated skin, it was postulated that HMGA2 is a marker of proliferating keratinocytes rather than transformation [133]. Li et al. found that HMGA2 nuclear translocation occurs as papillomas progress to cSCC [134]. All evidence considered, a role for HMGA2 in early tumorigenesis is well-supported. 

Cripto-1 is another oncofetal antigen expressed in KC tumors that is implicated in skin tumorigenesis. In embryonic and fetal tissues, Cripto-1 acts as a co-receptor for activin receptor heterodimers [135]. In coordination with Cripto-1, activin heterodimers transduce signals from transforming growth factor β (TGF-β) family signaling ligands such as Nodal, participating in the establishment of proper body axes and organogenesis [135]. Welss et al. detected Cripto expression in eight of ten primary human BCC tumors by RT-qPCR [136]. cSCC tumors did not express Cripto-1 [136]. However, only two SCC samples were used in this study. Shukla et al. investigated the expression and function of Cripto-1 during skin carcinogenesis. Treatment with TPA carcinogen induced Cripto-1 expression in normal murine epidermis. Benign papillomas treated with DMBA/TPA also exhibited elevated levels of Cripto-1 [137]. Staining was reduced in cSCC tumors [137]. Recombinant Cripto-1 treatment also stimulated proliferation and impaired differentiation of primary murine keratinocytes [137]. Mechanistic dissection revealed that Cripto-1 impairs TGF-β signaling, blocking the anti-neoplastic activity of TGF-β-induced senescence in Ras-transformed keratinocytes [137]. This work suggests that Cripto-1, like HMGA2, exerts considerable influence on non-transformed keratinocytes and is an important factor in early tumorigenesis.

These combined results indicate that ectopically-expressed developmental genes are significant contributors to BCC and cSCC pathogenesis (summarized in Table 2). We have presented evidence implicating ectopic developmental genes in multiple aspects of malignant progression, supporting future investigations to assess the robustness of the presented finding and to evaluate the possibility of their clinical translations.

## 4. Conclusions

This review highlights the contributions of CTAs and early developmental genes to KC pathogenesis. We discuss putative functions of ectopically expressed genes, based on studies of other malignancies, but not KCs. Recapturing the activity of these genes in a novel way can promote several hallmarks of cancer, including and not limited to: sustaining proliferative signaling, phenotypic plasticity and metabolic reprogramming. While this fact has been recognized in other cancers for some time, the contributions of CTAs and developmental genes to KC pathogenesis are only beginning to emerge.

Currently available chemotherapeutics, immunotherapies and targeted therapies for advanced/metastatic BCC and cSCC are wanting in efficacy and tolerability. Given the high prevalence of KCs, optimizing KC diagnosis and chemoprophylaxis are research priorities to minimize the prevalence and burden of disease. Hence, there is a persistent need to investigate the molecular pathogenesis of this cancer, and to develop new and improved approaches for disease management. Ectopically expressed CTAs and developmental genes are appealing to study as putative biomarkers and therapeutic targets, given their limited expression in healthy somatic cells.

Key questions remain. First, some genes that were classified as gametogenic or development-restricted are, in fact, detected in normal skin samples. Whether the expression of these genes in normal skin is a natural occurrence or a manifestation of environmental insults has implications for skin cancer prevention and early biomarker discovery. Analyzing differential expression of ectopically expressed genes between sun exposed versus non-sun exposed skin, and between young versus aged skin, can be an informative strategy to determine sensitive/specific biomarkers of early disease. In the same vein, characterizing CT and developmental gene expression in pre-malignant disease and carcinomas in situ can also yield novel insights and reveal therapeutic targets. In addition, despite recognized contributions to cancer biology, the majority of ectopically expressed CT, ESC and oncofetal genes have not been profiled in epidermal tumors. This remains a challenge that must be addressed. Even though expression for several genes has been detected in KC tumors, functional studies to assess possible contributions to disease phenotypes are lacking. Finally, there is the ever-present question of clinical translation. The pipeline from bench to bedside research is only emerging. Efforts to target and exploit CTAs and developmental gene expression for clinical uses are underway, and have occasionally manifested in concrete results such as clinical trials and biomarkers.

## Figures and Tables

**Figure 1 cancers-14-03630-f001:**
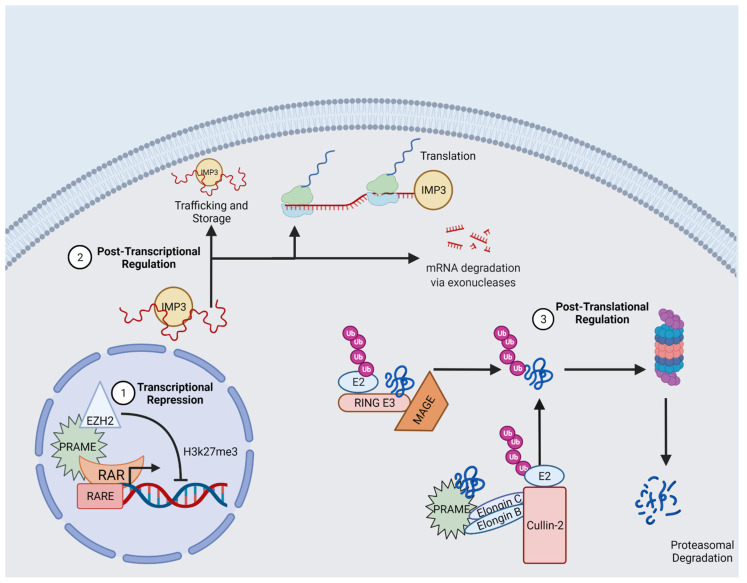
Ectopic CTA gene expression regulation at the transcriptional, post-transcriptional and post-translational levels. PRAME mediates transcriptional repression of retinoid signaling. IMP3 can regulate post-transcriptional regulation of gene expression, controlling mRNA trafficking and storage, translation and degradation. Both PRAME and MAGE regulate proteasomal degradation of certain substrates. Figure created using BioRender.com.

**Table 1 cancers-14-03630-t001:** Expression and functions of CTAs that are known to be expressed in KCs. * Description for all MAGE-A proteins are the same. +: positive expression. N.P: Not Profiled. AK: Actinic Keratosis.

Gene	Description	Expression in BCC	Expression in cSCC	Expression in Other Skin Tumors	Functions in Keratinocyte Carcinoma	References
A. GENE AND PROTEIN REGULATION						
PRAME	Repressor of retinoic acid signaling in gametes and embryonic stem cells Substrate-recognition subunit for Cullin-E3 ubiquitin ligase	+	+	N.P.	Associated with poorly differentiated tumors. Enriched in acantholytic SCC cells.	[41,50]
MAGE-A3	* Spatiotemporal localization and regulation of RING E3 ubiquitin ligases and substrates Protects the germline from environmental stressors	+	+	N.P.	Associated with PNI and poor differentiation in cSCC. Regulation of cyclins.	[41,68,69]
MAGE-A4	*	+	+	AK + Bowenoid AK+	PNI and poor differentiation in cSCC.	[41,66,68,69]
MAGE-A9	*	+	+	N.P.	Functions not investigated in KC.	[41]
MAGE-A10	*	+	+	N.P.	Functions not investigated in KC.	[41,67]
MAGE-A12	*	N.P.	+	N.P.	Proliferation and invasion in cSCC.	[70]
IMP-3	mRNA binding protein regulating mRNA localization, stability and degradation	+	+	N.P.	Proliferation and invasion in SCC.	[59,60,61]
B. INVASION AND STRESS RESPONSE						
SPAG9	c-Jun N terminal kinase interacting protein expressed on spermatocyte acrosome	+	+	N.P.	Functions not investigated in KC.	[83]
TSGA10	Mitochondrial biogenesis and organization	+	+	N.P.	Functions not investigated in KC.	[91]
TEX101	GPI anchored acrosomal protein for fertilization	+	+	N.P.	Functions not investigated in KC.	[76]
C. UNKNOWN SIGNIFICANCE						
TSPY1	Unknown functions	N.P.	+	N.P.	Functions not investigated in KC.	[68]
NY-ESO-1	Unknown functions	+	+	N.P.	Functions not investigated in KC.	[41]
SPATA19	Mitochondrial biogenesis and organization	+	N.P.	N.P.	Functions not investigated in KC.	[76]
ODF	Sperm tail structural protein	+	N.P.	N.P.	Functions not investigated in KC.	[76]

**Table 2 cancers-14-03630-t002:** Expression and function of developmental core ESC, EMT and oncofetal genes in KC tumors and other keratinocytic skin tumors. +: positive expression. −: negative expression; AK: actinic keratosis; BD: Bowen’s Disease; N.P.: Not Profiled.

Gene	Description	Expression in BCC	Expression in cSCC	Expression in Other Tumors	Notes	References
A. CORE PLURIPOTENCY CIRCUITRY AND RELATED FACTORS						
Sox2	Core pluripotency circuitry	+	+	+AK	Proliferation and invasion in BCC via SPRK-PI3K/AKT. Tumorigenesis, stemness, metabolic reprogramming in cSCC.	[103,104,105,106,109]
Pitx1	Limb development	N.P.	+	N.P.	Coordinates with Sox2 and p63 to repress Klf4-mediated squamous differentiation.	[107]
Oct-4	Core pluripotency circuitry	+	+	N.P.	Reversible skin dysplasia in murine models. Supports tumor formation and stemness. Promotes an ESC gene expression signature.	[103,111,112,113]
Nanog	Core pluripotency circuitry	+	+	N.P.	Increasing expression of type 2 deiodinase enzymes in BCC cells. Promotes DNA damage in cSCC.	[103,115,119,120]
Zfx1	Oct-4 co-factor	+	+	+AK + BD	Necessary for BCC tumorigenesis in PTCH1 null mice.	[122]
B. EMBRYONIC EMT TRANSCRIPTION FACTOR						
Zeb1	EMT	−	+	N.P.	Upregulated by thyroid hormones, which supports invasion and EMT in cSCC.	[131]
Twist1	EMT	+	+	N.P.	Stemness; Expression in keratinocytes regulates tumorigenesis and differentiation.	[128,129]
C. ONCOFETAL PROTEINS						
HMGA2	Chromatin binding protein	N.P.	+	N.P.	Induced by UVR in normal keratinocytes.	[133]
Cripto-1	Co-receptor for Nodal/GDF-1/2 signaling via activin receptor heterodimers	+	−	N.P.	Impairs anti-neoplastic TGF-b SMAD signaling.	[136,137]

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
