# Peer review of "The Contributions of Cancer-Testis and Developmental Genes to the Pathogenesis of Keratinocyte Carcinomas"

_cancers, 2022, doi:10.3390/cancers14153630_

Round 1

Reviewer 1 Report

This is well-written comprehensive and timely review on the role of CTAs and developmental genes in the pathogenesis of keratinocyte malignancies. A few minor errors need to be fixed. For example, the numbering of the subhead "1.2. CTAs May Influence Invasion and Stress Response Phenotypes in KC" is inverted.

The authors must be commended for assembling a comprehensive, timely and easily readable review.

 English language and style are fine/minor spell check required

Author Response

We wholeheartedly thank the reviewer for his or her helpful feedback. We have made the suggested changes in the resubmitted version of the paper. 

Sincerely, 

Dr. Litvinov 

Reviewer 2 Report

This review article summarized the contributions of CTAs and early developmental genes to KC pathogenesis. The authors performed comprehensive literature study and the manuscript was well written. I have a couple of minor comments:

1)    The first paragraph of the introduction section can be improved. In the manuscript, the authors did not discuss how de-differentiation, impaired differentiation and trans-differentiation contribute to KC tumorigenesis and development. In this paragraph, the authors do not need to describe details of the three subclasses of phenotypic plasticity, which was misleading for readers to understand the focus of the manuscript.

2)    Page 7, section “1.2.” should be “2.2”

Author Response

Thank you very much for your insightful comments and feedback. We have read and understood your feedback and have made efforts to integrate them into our work.

Thank you for making us aware of our typographic error on page 7 (incorrect numbering of subheading 2.1). We have corrected the error, and have reviewed other subheadings and references to ensure that they have been appropriately assigned. As requested, we have also increased the length of our simple summary.

Regarding Reviewer 1st criticism of the first paragraph, we agree that the discussion on phenotypic plasticity and the descriptions of the three subclasses was unnecessary and possibly misleading to readers from the outset. Thank you for drawing this to our attention. We have removed these descriptions and focused the first paragraph more generally on how ectopic gene expression occurs and how it can contribute to cancer pathogenesis. We still felt it is important to briefly mention how ectopic gene expression alters differentiation, since de-differentiation and trans-differentiation are mentioned later in the text. The discussion of phenotypic plasticity was meant to provide broader context and contemporary relevance to the study of ectopically-expressed genes in cancer. However, we acknowledge that the opening paragraph must be focused on introducing the scope of the article.

Warmest Regards,

Dr. Litvinov